# SmartSwim, a Novel IMU-Based Coaching Assistance

**DOI:** 10.3390/s22093356

**Published:** 2022-04-27

**Authors:** Mahdi Hamidi Rad, Vincent Gremeaux, Fabien Massé, Farzin Dadashi, Kamiar Aminian

**Affiliations:** 1Laboratory of Movement Analysis and Measurement, EPFL, CH-1015 Lausanne, Switzerland; kamiar.aminian@epfl.ch; 2Institute of Sport Sciences, University of Lausanne, CH-1015 Lausanne, Switzerland; vincent.gremeaux@unil.ch; 3Sport Medicine Unit, Division of Physical Medicine and Rehabilitation, Swiss Olympic Medical Center, Lausanne University Hospital, CH-1011 Lausanne, Switzerland; 4Gait Up S.A., CH-1020 Lausanne, Switzerland; fabien.masse@gaitup.com; 5Huma Therapeutics Ltd., London SW1P 4QP, UK; farzin.dadashi@huma.com

**Keywords:** sports biomechanics, swimming, IMU sensor, performance evaluation, feedback

## Abstract

Swimming coaches provide regular timed and technical feedback to swimmers and guide them efficiently in training sessions. Due to the complexity of swimmers’ performance, which is not visible in qualitative observation, quantitative and objective performance evaluation can better assist the coach in this regard. Inertial measurement units (IMUs) are used in swimming for objective performance evaluation. In this study, we propose a new performance evaluation feedback (SmartSwim) using IMU and investigate its effects on the swimmer’s weekly progress. Measurements were conducted each week with 15 competitive swimmers for 10 weeks using a Sacrum IMU. The SmartSwim report included a comprehensive representation of performance based on goal metrics of each phase extracted from the IMU signals. The swimmers were divided into two groups: the experimental and control groups. The SmartSwim report for each swimmer in the experimental group was given to the coach, who used it to adjust the training accordingly. The results showed that the experimental group outperformed the control group when comparing each swimmer, each session and the whole sessions. At the level of each individual, more members of the experimental group showed significant downward trend of average lap time (Mann-Kendall trend test, 95% confidence level). While comparing the sessions, the experimental group showed significantly lower lap time than the control group from the sixth session onwards (*p*-value < 0.05 from *t*-test). Considering all sessions, the experimental group showed significantly higher progress, lower average lap time, and more consistent records (Mann-Whitney U test at 95% confidence level) than the control group. This study demonstrated that SmartSwim can assist coaching by quantitatively assessing swimmers’ performance, leading to more efficient training.

## 1. Introduction

Swimming can be classified as a complex task because it cannot be mastered in a single session and has multiple degrees of freedom [1]. Learning such a complex physical activity and mastering the optimal technique for its execution depend on the continuous assessment of its performance. When it comes to complex tasks in sport, augmented extrinsic feedback has been shown to be necessary and effective for the athlete progress and development [2], regardless of the feedback modality. Therefore, the goal for successful coaching in swimming is clear: provide high-quality feedback concurrently or shortly after the activity on a frequent basis [3].

As in any other sport, swimming coaches rely mainly on their observations and coaching experience to monitor and evaluate swimmers’ performance. However, such subjective and qualitative analysis is not accurate enough to provide precise information about a swimmer’s strengths and weaknesses [4]. The complex nature of swimming has also led the research community to study it with new tools and systems from different perspectives, such as physiology [5,6], motor control [7], and biomechanics [8,9]. As a result, more attention has been paid to the use of sophisticated analytical systems by both researchers and coaches to obtain an objective and quantitative assessment of swimming performance [10]. Despite all the novel analysis methods that have been proposed for swimming analysis, there is a lack of an appropriate analysis system that can help both coaches and swimmers in better performance analysis [4]. Video-based systems, most commonly used as the gold standard in swimming, suffer primarily from shortcomings such as the time-consuming process of calibrating and digitizing landmarks, image distortion due to water reflections and air bubbles, and small capture volume in aquatic environments [11]. In contrast, ease of use, accessibility, easy-to-understand results, and feedback are the top four priorities of coaches in an analysis system [4].

In one of the oldest studies of feedback in swimming, Chollet et al. converted hydrodynamic pressure applied to the swimmer’s palm into auditory information. The swimmer was able to maintain stroke velocity, and improve motion stability and control through real-time sonification [12]. Visual feedback using a robot swimming under the swimmer for qualitative performance correction [13] or a complicated integrated system consisting of LED markers, a force plate, a high-speed video camera, an underwater camera, and a pressure pad for start, swimming and turn analysis [14] are examples of other studies using bulky systems to provide real-time feedback to swimmers. The use of such complicated sensor networks makes it difficult to use these systems in daily training. However, recent rapid improvements in the accuracy, size, and cost of inertial measurement units (IMUs) have made IMUs a credible option for swimmer motion tracking, as they can provide fast and easy-to-use feedback on detailed performance-related metrics [15].

Many studies have extracted kinematic variables from IMUs and shown them to be a powerful tool for swimming analysis [16,17,18], but some of them transmitted the results as feedback to the swimmer or coach. SwimMaster is a system based on three accelerometers at the wrist, lower back, and upper back that provides visual, tactile, and auditory feedback on average swim velocity, stroke time, and body orientation [19]. Rocha et al. used a network of five IMUs, a heart rate sensor, and a temperature sensor in a swimsuit to communicate information about the swimmer’s heart rate, stroke rate, and body temperature to the coach [20]. Silva et al. placed an IMU on the upper back of the swimmers to transmit information about the type of technique, laps, and strokes detection to the coach [21]. ISwimCoach is another analysis system that transmits to the coach the correct hand movement during strokes using a wrist IMU [22]. The system achieved 91% accuracy in detecting the correct strokes. Mangin et al., developed the idea of an instrumented glove that monitors hand movement during strokes and differentiates between recreational and elite swimmers using a wrist IMU [23]. According to the literature, the use of IMUs for feedback is still in its early stages. Although performed with a variety of variables, techniques, and modalities, researchers have focused mainly on the strokes of swim phase. These studies also led to numerous interferences in the normal swimming technique through a complex multi-sensor network. Moreover, the previous studies have rarely reached the field test to show the effect of feedback on swimmers’ performance.

Using the signals of a single sacrum-worn IMU, we developed a new approach in a previous study to segment a swimming lap into push, glide, strokes preparation, and swim phases [24]. Then, a phase-based performance evaluation was conducted to estimate goal metrics representing a swimmer’s performance in each swimming phase [25]. The objective of this study was to evaluate the in-field use of a comprehensive phase-based performance evaluation obtained from a single IMU as feedback to the coach. The goal metrics were shared with the coach to provide objective advice to swimmers in an experimental group and to adjust each individual’s training. We hypothesize that the objective feedback based on SmartSwim will improve the performance of experimental group to a higher degree compared to a control group that received routine feedback.

## 2. Materials and Methods

### 2.1. Measurement Setup and Protocol

Fifteen swimmers (9 males, 7 females, age: 14.6 ± 0.8 years, height: 171.6 ± 6.9 cm, body mass: 55.9 ± 10.1 kg) of a competitive team participated in this study. They had similar performance levels (50 m front crawl record: 28.60 ± 2.04 s) and were placed on the same team by the swimming club. The swimmers had similar training experiences and regularly trained together six days per week under the supervision and guidance of the same coach. A single IMU (Physilog^®^ IV, GaitUp, Lausanne, Switzerland) was attached to the swimmer’s sacrum with a waterproof tape (Tegaderm, 3M Co., USA) and recorded 3D angular velocity (±2000°/s) and 3D accelerometer (±16 g) at a sampling rate of 500 Hz. To make the sensor data independent of sensor placement on the swimmer’s sacrum, a functional calibration with simple movements (standing upright and squats) was performed before starting the test out of the water [18].

After a brief warm-up set by the coach, each swimmer completed five laps of one-way front crawl at maximum speed. Each participant had five minutes rest between two consecutive trials to avoid fatigue. Swimmers were asked to complete all swimming phases (push, glide, strokes preparation, swim) so that we could analyze their performance within each phase (Figure 1). Lap time was measured and recorded by the coach using a stopwatch for each lap during all test sessions. The average of the five lap times was used as their performance level. The same measurement was repeated once at the end of each week for ten weeks. The order in which the swimmers participated was the same in all sessions. The testing procedure was presented to each swimmer and they were asked to provide written informed consent prior to participation. The measurement protocol of this study was approved by the EPFL Human Research Ethics Committee (HREC, No. 050/2018).

#### Experimental and Control Groups

The swimmers were divided into two groups, an experimental and a control group. Since performance is assumed to be related to lap time as a key metric, the lap times of the first test session were considered as the baseline and used to select the swimmers of the two groups. The two groups were selected to have similar performance levels (as measured by lap time), similar age range, similar physical characteristics (body mass and height), and similar gender. The characteristics of the two groups are shown in Table 1. In this study, only for the experimental group, the coach received a comprehensive report from IMU as feedback, while the control group received only the usual feedback based on the coach’s observation.

### 2.2. SmartSwim Solution for Swimming Analysis and Feedback

The SmartSwim solution proposed in this study consists of two parts. In the first part, we performed a phase segmentation of each lap using our previously validated algorithms [24], and then estimated the goal metrics for performance evaluation in the different phases [25]. In the second part, we introduce a new feedback report based on these goal metrics to give the coach a comprehensive view of the performance and progress of each swimmer and the group.

#### 2.2.1. Phase-Based Performance Evaluation

Following our previous study evaluating swimming performance with a sacrum-worn IMU, each lap was segmented into the push, glide, strokes preparation, and swim phases [24]. The following goal metrics corresponding to each phase were estimated using a selection of kinematic variables by the data obtained from IMU [25]:Push phase: push maximum velocity.Glide phase: glide end velocity.Strokes preparation phase: strokes preparation average velocity.Total Swim phase: swim phase average velocity.Swim phase strokes: average velocity per stroke of the swim phase.Whole lap: lap average velocity.

The errors attributed to lap segmentation and goal metrics estimation are explained in the corresponding papers. Although this analysis was performed for both the experimental and control groups, only the reports of the experimental group members were given to the coach.

#### 2.2.2. Feedback Reports and Illustrations

For the experimental group, three types of feedback were given to the coach: (i) individual performance per session, (ii) individual multi-session performance, and (iii) comparison of swimmers per session. The reporting format was visually tailored to the coach’s needs to facilitate understanding and make it more efficient.

An example of the individual feedback provided after each session is shown in Figure 2. For each of the five laps (L1 to L5), a goal metric value was provided on each axis of a radar chart. In addition, the average and best performance in each phase for all five laps were added (Figure 2, right). In this type of representation, the pentagon of best performance is an imaginary lap that the swimmer can complete if he/she does their best in all swimming phases. In addition to the radar chart, a stroke velocity diagram was added to show the average velocity per stroke during the five laps (Figure 2, left). Furthermore, in this diagram, the stroke regularity can be observed by the variability of the inter-stroke velocity variability represented by the standard deviation values of each lap.

The individual multi-session result is the second type of feedback, including the swimmer’s average performance graphs during all previous sessions (Figure 3, left). The graph shows the swimmer’s progress in each goal metric during multiple sessions and indicates the percentage of change from the previous session at the bottom. The average lap time recorded by the coach for all previous sessions is also included in the report (Figure 3, right), allowing the coach to simultaneously observe the effect of the change in the goal metric on the lap time.

The third type of feedback per session is to compare the swimmers by plotting the average performance of each swimmer on the same radar chart (Figure 4). The coach can easily compare the swimmers at each phase and decide how to adjust the training for each individual, or design a specific training modification if all swimmers show the same weaknesses.

We shared the report of the experimental group’s performance with the coach. He considered the reports for each swimmer and adjusted the training sessions accordingly. We asked the coach to explain his observations and findings from the feedback and then write down the training changes planned for the next week for each swimmer. The charts for single-session and multi-session feedback were also explained and shared with the swimmers so they could self-monitor during the training sessions.

### 2.3. Feedback Effect Statistical Analysis

Because lap time is considered as the relevant measure of swimming performance, we evaluated the lap times of both groups for performance comparison. The two groups were compared at three levels: (i) per person, (ii) per session, and (iii) all sessions.

At the person-level, the trend of average lap times during the ten sessions for each swimmer was analyzed using the nonparametric Mann-Kendall trend test [26,27]. The purpose of this analysis was to determine how many swimmers in each group showed a significant trend of decreasing lap time due to performance progress. The test calculates the *S* value, which is the number of positive minus the number of negative differences when comparing all observations (Equation (1)).
(1)$$S=\overset{n-1}{\underset{j-1}{\sum }}\overset{n}{\underset{i-j+1}{\sum }}sgn\left(x_{i}-x_{j}\right)$$
where $x_{i}$ and $x_{j}$ denote the average lap time obtained in the *i*th and *j*th sessions, respectively, and n is the total number of sessions. For the populations with n≤40, it is sufficient to determine the corresponding probability of the Mann-Kendall trend test for the calculated value of *S* to find out whether the trend is significant or not. The trend significance for the swimmers with absences was analyzed for the existing number of records. A significance level of 95% is used for this analysis.

In the level of per session, we compared the lap times of the two groups in each session to determine if the experimental group significantly outperformed the control group. For this comparison, all lap time values for both groups in each session (40 values for eight swimmers in the experimental group and 35 values for seven swimmers in the control group) were compared. Since there is enough data for parametric test, first, the normality of the data distribution was checked using the Kolmogorov–Smirnov normality test [28] and then an independent t-test assuming unequal variances [29] was performed to compare the average values of the two groups, accepting a confidence level of *p*-value <0.05 as significant. The second analysis at this level is the comparison of the standard deviation of lap times in each session. For this analysis, the standard deviation of the five lap times for each swimmer was calculated, averaged across the group, and then compared to the other group. Because the sample size for this analysis is small (eight versus seven swimmers), the Mann-Whitney U test, a non-parametric method [30,31] with a 95% confidence level, was used for this comparison.

Finally, to compare the groups across all test sessions, the mean and standard deviation of the five lap times were estimated for each swimmer and then averaged across all swimmers in each group. The Mann-Whitney U test with a 95% confidence level was used to compare between groups across the ten sessions. Comparison of the means allows us to understand whether the experimental group was faster than the control group, and comparison of the means of the standard deviations evaluates the regularity of the swimmers between laps as a factor of efficient swimming. To test whether or not the overall progress of the swimmers with feedback was higher, we compared the average progress (average change in lap time compared to the first session) of the swimmers across all sessions using the Mann-Whitney U test.

## 3. Results

Four swimmers in the control group missed seven sessions due to swimmer unavailability, while swimmers in the experimental group participated in all test sessions. The average lap time of all swimmers during the ten sessions is shown in Figure 5. The swimmers of both groups show progress during the ten training sessions.

Comparing the first and the last session of all swimmers, the lap times of the last session are significantly lower than those of the first session, based on t-test results (*p*-value < 0.001 for both groups). For a qualitative comparison, the graphs for goal metrics and lap times of the first and tenth sessions are shown in Figure A1 and Figure A2 of Appendix A for the experimental group and the control group, respectively. On average, each swimmer of the experimental group improved 7.4% with respect to the first session while the control group swimmers improved 5.3%. However, the progress trend and also the amount of lap time change seem to be visually different for each swimmer. Moreover, the swimmers who had worse performance (higher average lap time) at the beginning of the measurements made more progress until the end of the measurements compared to others.

### 3.1. Coach Interpretation

Based on the reports the coach wrote during the measurements, he used feedback as an additional factor to his observation and Figure 6 can be conceptualized for his decision-making process with SmartSwim. In general, he relied on his experience and knowledge to make decisions. He considered the swimmer’s profile and existing constraints, such as time until the next competition or injury, to make a decision for each swimmer. SmartSwim provided the coach with new knowledge that enabled him to make safer and more reliable decisions to adapt each individual’s training.

A summary of the main comments and training adjustments mentioned in the coach’s reports can be found in Table A1 of Appendix B. By directly observing the individual performance evaluation chart (Figure 2, left), the coach identified the swimmer’s weaknesses in each session and evaluated the room for progress in each phase of swimming (by comparing the swimmer’s average and best performance). Multi-phase observation was helpful to the coach which is mentioned in coach comments. For example, when the swimmer started with a strong push (observed by push maximum velocity) but slacked off in the goal metrics of subsequent phases, the coach attempted to balance the performance between phases. Stroke average velocity chart (Figure 2, right) provided further information about the swim phase. The coach considered the results with little variation during strokes to be for the swimmers who can swim more regularly. He also qualitatively observed the effect of the change in swimming rhythm during breathing in the average velocity of the strokes. The coach also observed a decreasing velocity trend during strokes.

Multi-session feedback is used to monitor the effects of training on swimming performance over several weeks (Figure 3, left). Based on the coach’s comments, he assumed that any training adjustments would show an effect after three weeks. If he observed satisfactory progress, he continued training in the same manner; otherwise, he chose a different strategy for the swimmer. In addition, by looking at the lap time values in the same graph (Figure 3, right), the coach was able to observe the effect of the training adjustment on the swimmer’s average lap time and make a more reliable decision. As the final feedback type, the swimmer comparison chart (Figure 4) allowed the coach to see the weaknesses and strengths of each swimmer compared to the others. The coach also used this chart to find the swimmers with higher potential to focus on, as the swimmers’ progress compared to others was clearly visible when looking at this chart over several sessions.

### 3.2. Statistical Analysis

The results of the applied Mann–Kendall trend test for the person-level comparison (Table 2) show a decreasing trend in lap time for swimmers in both groups. In the experimental group, this trend was significant for all but one swimmer, while in the control group only two swimmers showed a significant trend. In addition, stronger significance (higher *S* values) was observed for swimmers in the experimental group.

At the session level comparison, after confirming the normality of the data by Kolmogorov–Smirnov test, an independent t-test was performed for comparing the groups. The two groups showed a significant difference from the sixth session onward (Figure 7). The standard deviation results of the groups also showed a significant difference from the sixth session (except for the eighth session) (Table 3).

Finally, the average lap times of the groups in all 10 test sessions are compared. Although both groups showed a significant decreasing trend in lap time (significant trend from Mann–Kendall test in Figure 7), the experimental group scored significantly lower lap times compared to the control group based on the Mann–Whitney U test with a confidence level of 95% (*U_stat_* = 18, *n*_1_ = *n*_2_ = 10, *p*-value < 0.05, two-tailed). The standard deviation of the experimental group is also significantly lower than that of the control group (*U_stat_* = 21, *n*_1_ = *n*_2_ = 10, *p*-value < 0.05, two-tailed). By taking the first sessions as the baseline, we quantified the progress of each swimmer as the difference between the lap time of each session and the baseline. According to the result of the Mann–Whitney test, the average progress of the members of the experimental group was significantly higher than that of the control group (*U_stat_* = 21, *n*_1_ = 8, *n*_2_ = 7, *p*-value < 0.05, two-tailed). Considering all swimmers in each group, the swimmers in the experimental group and the control group achieved an average progress of 0.65 s (4.4%) and 0.35 s (2.3%), respectively.

Since lap average velocity is the division of pool length by lap time, it is expected to correlate with progress. Therefore, we performed the same analysis in session level for lap average velocity to see if it showed the similar difference between the two groups. The results are explained in Appendix C (Table A2 and Figure A3). Similar to lap time, the average and standard deviation of lap average velocity becomes significantly different between the two groups after sixth test session.

## 4. Discussion

In this study, we proposed a new approach for performance evaluation and feedback. We investigated the effects of training with SmartSwim on the performance of a group of swimmers during a 10-week training period. The results obtained confirmed the ability of SmartSwim to provide objective feedback during the training sessions with a lightweight and portable IMU. After each test session, this feedback was communicated to the coach through a comprehensive report that illustrated the main goal metrics of the test to help the coach design the training more efficiently until the next measurement. Comparison with a control group that received only the usual feedback confirmed our hypothesis that the experimental group achieved better progress in terms of target performance (i.e., lap time) when they received advice based on performance-related goal metrics.

### 4.1. Using Feedback for Training

Based on the evaluations of the coaches’ reports, he quickly understood how to use SmartSwim feedback. Unlike traditional methods, he could easily identify and focus on the phase with a higher chance of progress, while it takes more time to determine a swimmer’s potential through subjective observation. He also observed the interaction between swimming phases and examined how well the swimmer managed to improve all phases together. Finding the optimal training for each swimmer requires trial and error [32], which is reduced when the coach has a quantitative assessment of performance. The coach mentioned that he had more confidence in leading the swimmers in the experimental group than in the control group. He found that the feedback was consistent with his personal judgement and followed the swimmers with numbers rather than pure observation. Using the graph of strokes average velocity, the coach recognized the skilled swimmers with low velocity variation [33], detected the breathing effect on swimming rhythm [34] and identified the swimmers’ endurance level [35]. The swimmers in the experimental group also received the feedback and review of their weekly progress with great enthusiasm, as none of them missed a single testing session.

In this study, the SmartSwim feedback report functioned as an assistant to the coach. The involvement of the coach is essential because the final decision to optimize training depends on the coach’s judgment and the swimmer’s profile. The coach usually relies on his or her personal experience, based on which he or she can usually make a qualitative assessment of the training sessions [4]. Our results show that objective and quantitative goal metrics complement the coach’s qualitative observations and allow him to better personalize his advice and test different strategies using the same goal metrics. Compared to similar studies [22,23], we were able to provide feedback on all phases of swimming, not just the swim phase, allowing the coach to obtain a more comprehensive assessment. Moreover, the feedback was tested in-field for the evaluation of its effect on real training sessions of the team.

### 4.2. Experimental and Control Group Comparison

Starting with the person-level comparison, the results of Mann–Kendall trend test with 95% confidence level showed that the decreasing trend in lap time during the ten training sessions is significant for seven out of eight swimmers in the experimental group, while only two swimmers in the control group showed such a significant trend. According to the logic of the Mann–Kendall trend test, a significant trend exists when the lap time decreases continuously from week to week. This confirms that although the swimmers in both groups achieved a lap time at the end of the ten weeks that was significantly lower than the baseline time, the swimmers in the experimental group made continuous progress during the measurements, which is a crucial factor in efficient training [36]. In addition, the S-values calculated for the swimmers in the experimental group (a range of [−41, −23]) are consistently lower than those of the control group (a range of [−24, −17]), reflecting the stronger improvement trends when using SmartSwim for coaching.

For the session-level assessment, the two groups had a similar lap time average and standard deviation in the first session (*p*-value > 0.05). During the training sessions, the coach trained the experimental group based on feedback, while for the control group he relied only on his own coaching experience. Consequently, the experimental group’s progress in the remaining sessions is influenced by feedback-based training. The results show that the average lap time and standard deviation of the experimental group are significantly lower than those of the control group from the sixth session (*p*-value < 0.05). This shows that the swimmers in the experimental group performed not only faster, but also more consistent and systematic [37] than the control group.

Focusing on the swimmers’ weaknesses and comparing personal observation with feedback helps the coach sharpen his critical thinking skills [38]. The coach provided more relevant and personalized feedback to each swimmer, which was reflected in higher progress of these swimmers during the same period compared to the control group. In addition, based on the results of using lap average velocity to perform the session-level assessment shown in Table A2 and Figure A3 of the Appendix C, similar results were obtained and the two groups differed significantly from the sixth session. The coach can use the lap average velocity as a substitute for lap time and focus on other goal metrics during the training sessions.

Finally, swimmers in the experimental group show lower lap time and standard deviation and higher progress when all test sessions are considered together (Mann-Whitney U test, *p*-value < 0.05). This suggests that the effect of feedback can be observed not only for each swimmer and session, but also in the overall picture of long-term training. Considering the importance of seasonal evaluations of swimmers [39], this level of comparison helps the coach to monitor the performance in entire season and better prepare for competitions. In summary, the superiority of the experimental group over the control group is evident when comparing the swimmers’ performance at three levels: per person, per session, and all sessions. Bielec et al. examined the effect of a specific aerobic exercise on the performance of young swimmers and found a significant improvement in males over two months of training [40].

We tried to keep all effective factors the same for the experimental and control groups, but we cannot claim that the superiority of the experimental group over the control group is solely due to feedback. The effects of factors such as psychology [41], nutrition [42], or physiology on swimmers’ performance were not considered in this study. Our study is limited in terms of the number of swimmers, so we had to use non-parametric analyses. A larger data set is needed to more conclusively evaluate the effect of feedback. We mixed male and female swimmers, regardless of their growth and maturation status, in the two groups and their individual comparison is beyond the scope of this study. Among the four main swimming styles, front crawl was analyzed, while the same feedback can be given for the other swimming styles. Lap times are recorded by coaches using a stopwatch, which is a source of error in our measurements. Due to technical limitations, we need to extract the data and analyze it before giving feedback to the coach which takes a few hours. However, the coach can compare his observations with the feedback report if it is provided during or shortly after each lap. SmartSwim also demands the swimmer to perform all swimming phases in sequence, not starting from the middle of a phase.

The main disadvantage of wearables is the increased water drag on the swimmer’s body [43]. The use of a single sensor in SmartSwim minimizes this problem and inconvenience to the swimmer. In addition, it does not interfere with the swimmer’s normal performance and can be used during daily training. The use and attachment of the sensor requires extremely little preparation and analysis on the part of the coach, who can therefore easily use the system for all swimmers at the same time. The sensor could be integrated and industrialized into the swimsuit at a later stage. Regarding the complexities of finding the best coaching approach for young swimmers, multiple studies examined the effect of training load [44], mental training [45], or training intensity [46] on the performance of young swimmers, rarely reporting significant performance changes. Since technique analysis is of high importance for efficient coaching, training program can be improved by SmartSwim feedback. Sharing the phase-based feedback of a larger group of swimmers with the coach and developing the appropriate real-time algorithms to provide feedback simultaneously can be offered as next steps in this research.

## 5. Conclusions

In this study, we examined the effects of coaching with SmartSwim, a new phase-based performance evaluation feedback, on swimmers’ performance during 10 weeks of training. The coach used a comprehensive report of phase-based goal metrics from IMU as an assistant for eight swimmers in the experimental group and adjusted their training accordingly, while he guided seven swimmers in the control group based only on his observations and coaching experience. The results showed that the experimental group outperformed the control group when considering the performance of each swimmer, the performance of the group in sessions, and the group performance in all training sessions. Most of the swimmers in the experimental group showed a significant downward trend in their average lap times in 10 test sessions. The experimental group significantly outperformed the control group in terms of lap times from the sixth session onward. In addition, the swimmers in the experimental group showed more consistent results than those in the control group. Finally, considering all 10 sessions, the swimmers in the experimental group showed significantly higher progress, lower average lap times, and more consistent records than the control group. The coach found the feedback reports very helpful in “diagnosing” the swimmers’ weaknesses and monitoring their progress more efficiently during the training sessions. This study has helped meet the needs of the coaching community and promote objective coaching in swimming.

## Figures and Tables

**Figure 1 sensors-22-03356-f001:**
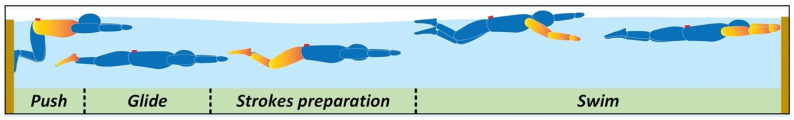
Measurement protocol. The swimmer starts in the water with a wall push-off and performs all swimming phases of push, glide, strokes preparation, and swim. The coach records the lap time with a stopwatch, while the phase-based goal metrics were extracted from the IMU (red box) worn on the sacrum.

**Figure 2 sensors-22-03356-f002:**
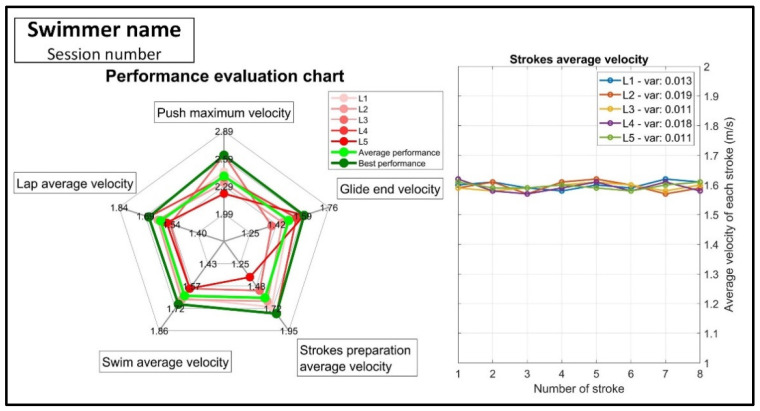
Individual feedback for the swimmer after the test session. The performance evaluation chart (**left**) shows the goal metrics for five laps, the average performance (light green) and the best performance (dark green). The stroke average velocity chart (**right**) shows the average velocity per stroke during five laps and its variation (“var”, corresponding to standard deviation) in the legend.

**Figure 3 sensors-22-03356-f003:**
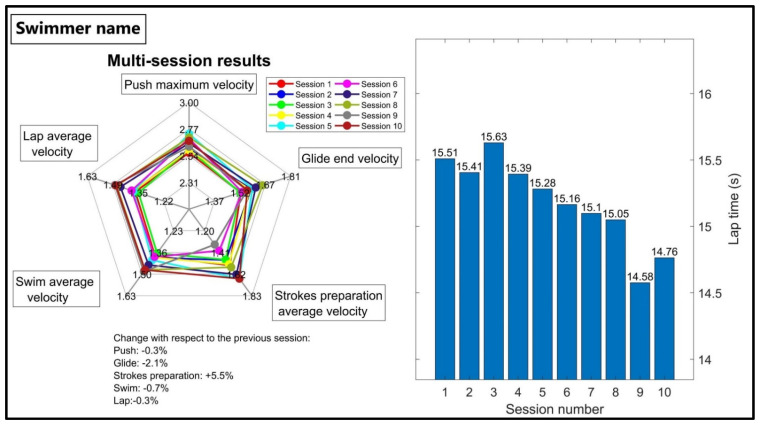
Feedback on multi-session performance evaluation feedback. The radar chart shows with a different color the average performance of all sessions (**left**) with changes compared to the previous session. The bar graph shows the average lap time of all sessions recorded by the coach (**right**).

**Figure 4 sensors-22-03356-f004:**
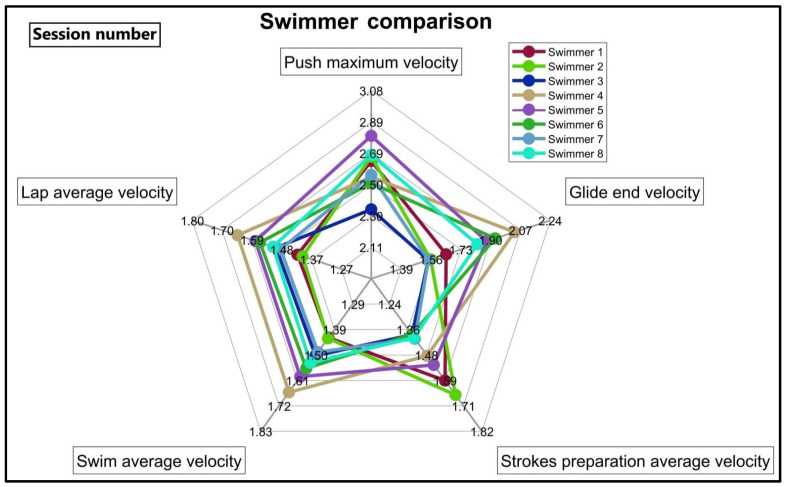
Feedback to compare the swimmer’s average performance in different swimming phases. The coach can see the strengths and weaknesses of each swimmer through this comparison.

**Figure 5 sensors-22-03356-f005:**
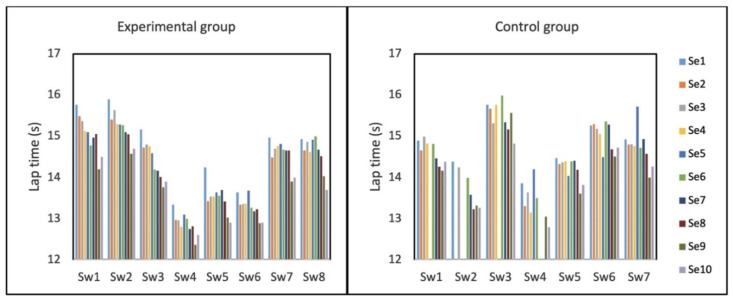
Average lap time of the swimmers in 10 sessions (Se1–Se10), for eight swimmers of the experimental group (Sw1–Sw8, (**left**)) and seven swimmers of the control group (Sw1–Sw7, (**right**)).

**Figure 6 sensors-22-03356-f006:**
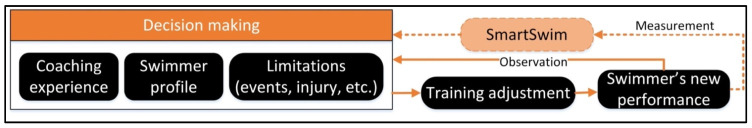
Feedback effect on the training procedure illustrated by the coach.

**Figure 7 sensors-22-03356-f007:**
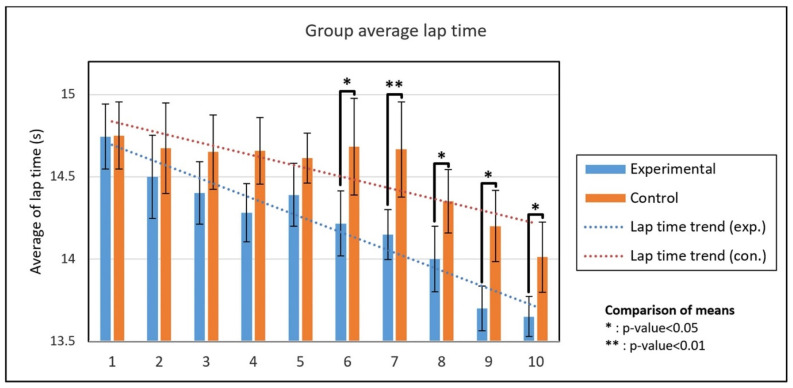
Average and standard deviation of lap times for the experimental and control groups during ten test sessions.

**Table 1 sensors-22-03356-t001:** Characteristics of the swimmers in the experimental and control groups. The swimmers were selected to have similar characteristics.

Group	Male	Female	Age (yrs)	Height (cm)	Body Mass (kg)	First Session Record in Seconds (Baseline)
Experimental	4	4	14.5 ± 0.5	170.1 ± 6.5	55.5 ± 8.3	14.74 ± 0.87
Control	4	3	14.6 ± 0.4	171.2 ± 7.1	54.9 ± 7.2	14.75 ± 0.79

**Table 2 sensors-22-03356-t002:** Person-level comparison between the experimental group and the control group. *S* value for the Mann-Kendall trend test of average lap time values during 10 training sessions. A negative sign indicates a decreasing trend.

Experimental Group Lap Time Trends—*S* Value													
Sw1	Sw2		Sw3		Sw4		Sw5		Sw6		Sw7		Sw8
−37 *	−41 *		−39 *		−31 *		−19		−29 *		−25*		−23 *
**Control Group Lap Time Trends—*S* Value**													
Sw1		Sw2		Sw3		Sw4		Sw5		Sw6		Sw7	
−24 *		−17 *		−16		−14		−18		−15		−23 *	

* Significant with 95% confidence level.

**Table 3 sensors-22-03356-t003:** Session-level comparison between the experimental and control groups. t score and U score results for comparison of mean and standard deviation lap times, respectively.

Test Session	1	2	3	4	5	6	7	8	9	10
Lap time comparison: *t* score	0.27	1.62	1.36	1.81	1.12	2.39 *	2.79 **	2.09 *	2.40 *	1.99 *
Standard deviation comparison: U score	25	18	17	16	10	9 *	7 *	13	5 *	9 *

* *p*-value < 0.05, ** *p*-value < 0.01.

## Data Availability

The data presented in this study are available upon request from the corresponding author.

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
