# Peer review of "SmartSwim, a Novel IMU-Based Coaching Assistance"

_sensors, 2022, doi:10.3390/s22093356_

Round 1
Reviewer 1 Report
This manuscript presents an interesting performance evaluation feedback (SmartSwim) based on inertial measurement unit (IMU) to study its influence on the swimmer's weekly progress. This manuscript studied the progress of 15 competitive swimmers for 10 weeks employing a Sacrum IMU. The SmartSwim can help to improve the efficient training of swimmer's performance. This manuscript can be enhanced considering the following comments:
1.-Which are the main limitations or challenges of the SmartSwim?
2.-The resolution of all the figures should be improved.
3.-What parameters could affect the measurements obtained using SmartSwim?
4.-What is the future research work?
Author Response
Dear Reviewer,
Thank you for your comments. We did our best to address your points in the manuscript and improve it. You can find attached a word file with our answers to your questions and the corresponding changes in the manuscript.
"Please see the attachment."

Reviewer 2 Report
Dear authors,
I have carefully reviewed the manuscript entitled “SmartSwim, a novel IMU-based coaching assistance”, and I would like to congratulate you for the effort to carry out this study, in a clear attempt to bridge the gap between sport sciences and the real world of the sportive context, which addresses a relevant question regarding training monitoring, with direct and on-time applications for coaching decision making. In my opinion the overall appreciation of the manuscript is quite positive, considering the proper rationale and framework for the research question, the suitable methods and procedures followed, and the observed outcomes in accordance with the study purpose. Nevertheless, I have some concerns regarding the scarce and unexplored discussion of the results, that I seriously believe that can be overcome by the authors.
Abstract
The study purpose is not clearly presented. Please change, in order to make it perfectly objective and understandable to the potential reader.
Lines 21 – 22: This sentence is somewhat confusing. Please reformulate.
Line 28: “…SmartSwim assist coaching …”. Please add “can assist”
Introduction
The authors provide a proper framework and rationale for the understanding of the present study relevance.
Line 37: “…for the athlete progress and development”. Please change.
Line 80: “…during the strokes…”. Please remove “the”.
In the last paragraph I recommend authors to replace the first sentence to line 96, after presenting the previous studies using the SmartSwim [24, 25] and possible uses of these units.
Materials and Methods
Was growth and maturation assessed? If possible, more information regarding the sample characteristics should be provided. For example, I believe that at this competitive level or age-group, performance changes are more pronounced and significant, unlike in older athletes, where performance changes are less obvious. Nevertheless, and even if there is no significant decreasing trend, it would still be worthy to enhance swimming performance.
Line 144: Please insert reference here.
Validation and reliability values of the SmartSwim should be included in Line 152.
Procedures and statistics seems to be appropriate for the ibjective.
Results
Figure 5: Please correct order – Experimental group (8 swimmers) on the left, and Control group (7 swimmers) on the right, according to figure.
Lines 282 – 286: It is not common the use of references in the results section, particularly when interpreting and making inferences of the results, which is more typical in the discussion section.
Line 313: Please change the order, according to the element’s presentation, i.e., Lines 327 – 331 should be presented here.
Line 335: “According to the results of the Mann-Whitney…”. Please change.
Line 339: Could you provide the percentage differences please?
Discussion
In the discussion section authors do not use the available literature to support the major findings. This section is limited and very descriptive about the study results.
Line 376: This is a major finding and needs further exploration.
Line 378: “swimmer-level” – I suggest changing because swimmer are the same level and age-group, but different experimental conditions were applied.
Line 391: 6th – upper line. Please change.
Line 409: Growth and maturation characteristics were also not included, and this limitation should be recognized.
Line 446: Please insert a punch sentence, like the authors did in the abstract.
Author Response

(The authors gave the same response as above.)

Reviewer 3 Report
This paper reports a work about a training strategy for swimmers based on the speed signals from a IMU. The results show that a favorable promotion is achieved. The work may be very helpful in training the swimmers, but this paper is not very proper for the scientific journals. The paper is more like a specification or project report for the strategy “SmartSwim”. No theoretical importance or technical value can be found in this paper. So, in my opinion, this paper is not publishable for this journal.
Author Response

(The authors gave the same response as above.)

Round 2
Reviewer 2 Report
Dear Authors,
I would like to thank you for considering the reviewers suggestions. All the comments were addressed and integrated in the newest version. I feel that the resubmitted manuscript fits the highest standards of this journal, although a further and deeper exploration of the outcomes and practical applications regarding what is known in literature, coud be included.
Again, I would like to congratulate you on the effort to bridging the gap between science and the real world in sport, highlihting the importance of sport sciences in athletes performance enhancement and development.
Author Response
Dear Reviewer,
Thank you very much for your comments. You helped us improve the manuscript a lot. Regarding your comments in the second round, "Please see the attachment".
Best,

Reviewer 3 Report
Firstly, I fully understand the value and importance of this work. My concerns mainly focus on the conformity of this paper, which is considered by the academic editors. Now, the editors have approved this issue.
Then, based on the revised version and authors’ responses to other reviewers, I think the paper has been improved, making it suitable for publication.
Author Response
Dear reviewer,
Thank you for your comments.